# AlH₃ as High-Energy Fuels for Solid Propellants: Synthesis, Thermodynamics, Kinetics, and Stabilization

Youhai Liu [1], Fusheng Yang [1,*], Yang Zhang [2], Zhen Wu [1] and Zaoxiao Zhang [1]

[1] School of Chemical Engineering and Technology, Xi'an Jiaotong University, Xi'an 710049, China; 4121116019@stu.xjtu.edu.cn (Y.L.); wuz2015@mail.xjtu.edu.cn (Z.W.); zhangzx@mail.xjtu.edu.cn (Z.Z.)
[2] Science and Technology on Combustion and Explosion Laboratory, Xi'an Modern Chemistry Research Institute, Xi'an 710065, China; aerozy@163.com
[*] Correspondence: yang.fs@mail.xjtu.edu.cn

**Abstract:** Aluminum hydride (AlH₃) has attracted wide attention due to its high gravimetric and volumetric hydrogen capacity. AlH₃ can easily release hydrogen when heated at relatively low temperature. Such high hydrogen density and low dehydrogenation temperature make it one of the most promising high-energy fuels for solid propellants. In particular, AlH₃ as a component of solid propellants may greatly increase the specific impulse of rocket engines. However, AlH₃ exhibits low chemical and thermal stability in an ambient atmosphere. In this paper, the research progress about the synthesis, dehydrogenation thermodynamics, and kinetics, the stabilization of AlH₃ over the past decades are reviewed, with the aim of exploring more a economical synthesis and suitable stabilization methods for large-scale use in solid propellants. Finally, some suggestions regarding future research directions in this filed are proposed.

**Keywords:** AlH₃; solid propellants; decomposition; synthesis; stability; high energy material





## 1. Introduction

Solid propellants act as the power source of the rocket motor and play an important role in the fields of military and aerospace. The pursuit of higher energy solid propellants has always been an inevitable requirement for the development of solid rockets. Metal hydrides possess high combustion heats due to the introduction of "hydrogen energy", thus making them promising candidates for the solid fuels of propellants. In addition, considering that a decrease in the average molecular weight of gas combustion products will lead to an increase specific impulses of propellants, the low-molecular-weight H₂ produced by hydride decomposition makes them more attractive.

In recent years, considerable efforts have been made to evaluate the potential of hydrides as energy fuels for solid propellants, including aluminum hydride (AlH₃) [1–3], lithium hydride (LiH) [4], magnesium hydride (MgH₂) [5,6], titanium hydride (TiH₂) [7,8], and ammonia borane (NH₃BH₃) [9]. Table 1 summarizes the hydrogen storage properties of common metal hydrides. Among the above hydrides, AlH₃ is regarded as an important promising candidate due to its high heat of combustion (about 40 MJ/kg), volumetric hydrogen storage densities (148 kgH₂/km³) and hydrogen content (10.1 wt%). For example, when replacing 18% of aluminum (Al), a widely used metallic fuel in solid propellants, in a conventional ammonium perchlorate/hydroxyl-terminated polybutadiene (AP/HTPB)-based rocket propellant with aluminum hydride (AlH₃), the theoretical specific impulse would increase by 10%, and the flame temperature would decrease by 5%.

AlH₃ is a polyphasic substance, with no less than seven non-solvated crystal forms, namely α, α', β, γ, δ, ε, and ζ-AlH₃, due to different synthesis routes and reaction conditions. It is generally believed that the α-AlH₃ phase is the dominant phase, while the other phases are usually considered as impurity products of the α-AlH₃ phase, and they are not as stable as α-AlH₃. Some of them directly decompose into elements, while the other ones

convert to $\alpha$-AlH$_3$ before further decomposition. Therefore, $\alpha$-AlH$_3$ deserves much more investigation [10,11]. However, $\alpha$-AlH$_3$ is very sensitive to oxidants and could easily decompose into aluminum in an atmospheric environment [12]. Due to the poor stability of AlH$_3$, during long-term storage process, the slight decomposition of AlH$_3$ at room temperature promotes the generation and development of cracks in the solid propellant grain and reduces the propellant performance. In addition, the poor compatibility of $\alpha$-AlH$_3$ with certain propellant components, such as nitrates, further limits its application in solid propellants [13].

**Table 1.** The hydrogen storage properties of common metal hydrides.

| Hydrides | Molar Mass (g/mol) | Density (g/cm) | Gravimetric Hydrogen Density (wt%) | Volumetric Hydrogen Density (kg/m$^3$) | $T_{dec}$ (K) |
|---|---|---|---|---|---|
| LiH | 7.95 | 0.82 | 11.5 | 98.3 | 474.15 |
| MgH$_2$ | 26.31 | 1.45 | 7.6 | 110 | 674.15 |
| TiH$_2$ | 49.89 | 3.91 | 4.0 | 91.0 | 624.15 |
| NH$_3$BH$_3$ | 30.81 | 0.78 | 19.6 | 145 | 399.15 |
| BeH$_2$ | 11.03 | 0.65 | 18.28 | 71.2 | 524.15 |
| AlH$_3$ | 30.0 | 1.477 | 10.1 | 148 | 434.15 |

A large quantity of endeavors have been tried on the synthesis and stabilization of $\alpha$-AlH$_3$ [14–20], aiming at producing products with better stability, higher quality, and more energy as additive fuel for solid propellants. This review focuses on the relevant research about the above-mentioned aspects; meanwhile, thermal decomposition properties and stabilization methods were also proposed. The paper is organized as follows. Firstly, Section 2 presents the synthesis methods of aluminum hydride. Next, the thermodynamics and kinetics of AlH$_3$ are described in Section 3. Thirdly, Section 4 summarizes the physical and chemical modification methods of AlH$_3$ to achieve the purpose of stabilizing aluminum hydride. Finally, we propose potential future research directions and aim to offer a valuable reference for subsequent studies on AlH$_3$ in related fields.

## 2. Synthesis of AlH$_3$

### 2.1. Wet Chemical Methods

Synthesis of AlH$_3$ has been studied since the 1940s, when Stecher and Wiberg first synthesized AlH$_3$·2N(CH$_3$)$_3$ in a low and impure form. Later, wet chemistry method was first reported by Finholt et al. [21], but the ether cannot be completely removed at that time, so the products were mainly solvated $\alpha$-AlH$_3$.

$$4LiH + AlCl_3 \xrightarrow{ether} LiAlH_4 + 3LiCl \downarrow \tag{1}$$

$$3LiAlH_4 + AlCl_3 \xrightarrow{ether} 4AlH_3 + 3LiCl \downarrow \tag{2}$$

Brower et al. [22] synthesized seven non-solvated AlH$_3$ polymorphs, which are now commonly referenced when preparing AlH$_3$. The process consists of two steps. Firstly, LiAlH$_4$ (or LiH) and AlCl$_3$ react in ether solution to form AlH$_3$–ether complex and LiCl precipitate, seen in Reaction (1). Secondly, the ether ligand was removed through heating the product, which is also called post-crystallization, as shown in Reaction (2). Finally, the products need to undergo an ether solvent wash to eliminate any surplus LiAlH$_4$ or LiBH$_4$, followed by vacuum drying to produce the final AlH$_3$.

$$3LiAlH_4 + AlCl_3 + nEt_2O \rightarrow 4AlH_3 \cdot nEt_2O + 3LiCl \downarrow \tag{3}$$

$$AlH_3 \cdot nEt_2O \xrightarrow{time,\ temperature,\ LiAlH_4,\ LiBH_4} AlH_3 + nEt_2O \uparrow \tag{4}$$

According to Refs. [23–26], the synthesis process of the AlH$_3$ polymorphs depends on multiple conditions, such as temperature, time, the presence of LiAlH$_4$/LiBH$_4$ or not, and the atmosphere. Generally, the γ-AlH$_3$ can be obtained from the desolvation reaction of etherate AlH$_3$ in the presence of only LiAlH$_4$ at lower temperature (60∼70 °C), whereas the α-AlH$_3$ can be obtained from the desolvation reaction of etherate AlH$_3$ in the presence of excess LiBH$_4$ and LiAlH$_4$. Pure β-AlH$_3$ or α'-AlH$_3$ are difficult to prepare and usually form accompanying with γ-AlH$_3$. However, no reproducible synthesis methods of the δ-, ε-, and ζ-AlH$_3$ were established.

Table 2 displays the preparation conditions and properties of various AlH$_3$ polymorphs. Additionally, a more efficient method for synthesizing high-purity α-AlH$_3$ crystals involves directly dissolving the reactants in ether and toluene, respectively.

**Table 2.** The preparation conditions and characteristics of different AlH$_3$ polymorphs.

| Polymorphs | Experiment Conditions | | | Size of Product | Structure | Space Group | Morphology |
|---|---|---|---|---|---|---|---|
| | LiAlH$_4$:AlCl$_3$:LiBH$_4$ | T (°C) | Time | | | | |
| α-AlH$_3$ | 1:4:1 | 65 | 6.5 h | 60 nm | a = 4.449 Å c = 11.804 Å | $R\overline{3}c$ |  |
| | 1:3:0 (PDMS, HCl) | 85–93 | 2–8 h | 6–13 μm | | | |
| | 1:4:0 (γ → α) | 62 | 11 h | 50–100 μm | | | |
| | 1:4:1(β→α) | 65 | 6 h | | | | |
| α'-AlH$_3$ | 1:4:0 | 60 | 4 h | 1 μm | a = 6.470 Å b = 11.117 Å c = 6.562 Å | *Cmcm* |  |
| β-AlH$_3$ | 1:4:1 | 75 | 6 h | <50 μm | a = 9.004 Å | $Fd\overline{3}m$ |  |
| γ-AlH$_3$ | 1:4:0 | 65 | 45 min | <50 μm | a = 7.336 Å b = 5.367 Å c = 5.765 Å | *Pnnm* |  |

For wet synthesis methods, the formation of AlH$_3$·Et$_2$O requires a large amount of environmentally harmful organic solvents, like ethers or amines. Furthermore, the costly removal and recycling of these organic solvents, coupled with the challenge of controlling the preparation conditions due to the reaction's high sensitivity to temperature and time, make achieving the desired polymorph difficult [27]. Conditions control, solvent reuse, and other factors make it relatively difficult to scale up the synthesis of AlH$_3$ using wet methods.

## 2.2. Dry Synthesis Methods

As mentioned above, wet synthesis of AlH$_3$ requires a large amount of solvents, often being expensive and toxic. Therefore, numerous methods for the synthesis of AlH$_3$ without solvents, generally termed dry synthesis methods, have been proposed [28–34].

### 2.2.1. Mechano-Chemical Method

The mechano-chemical method is considered more environmentally friendly and cost-effective than traditional wet chemical methods, and has been used to prepare numerous aluminum hydrides, such as Mg(AlH$_4$)$_2$ [35,36], Ca(AlH$_4$)$_2$ [37,38], and LiMg(AlH$_4$)$_3$ [39]. Many studies have demonstrated that this solvent-free method has lower energy consumption than traditional wet synthesis process [30,31], and only necessitates inexpensive metal hydrides (such as LiH, NaH, CaH$_2$, and MgH$_2$) instead of LiAlH$_4$, making it an increasingly popular alternative method for AlH$_3$ preparation.

Duan et al. [29,40] investigated the synthesis of AlH$_3$ by reactive milling using aluminum chloride and cheap metal hydrides as starting materials. XRD and NMR analyses indicated that nano-sized materials AlH$_3$ were prepared using commercial AlCl$_3$ and nanocrystalline MgH$_2$ as reagents.

Duan et al. [30,41] also reported that α-AlH$_3$ nanocomposites can be obtained by adding TiF$_3$ to LiH and AlCl$_3$ systems through ball milling in a short period of time, where suitable gas and pressure conditions can completely suppress the formation of metallic Al. The results indicate that TiF$_3$ plays a synergistic role in the solid-state reaction between LiH and AlCl$_3$, and has significantly affect dehydriding properties of AlH$_3$, compared to the α-AlH$_3$/LiCl nano-composite without TiF$_3$. The as-milled α-AlH$_3$/LiCl-TiF$_3$ composite has a hydrogen desorption of 9.92 wt% at 160 °C within 750 s, which is very close to the theoretical hydrogen capacity of AlH$_3$.

However, this method is not suitable for large-scale commercial applications due to prohibitively high energy consumption. In addition, it's hard to get a pure product since slight decomposition of AlH$_3$ is almost inevitable during process.

### 2.2.2. Organoaluminum Decomposition Method

Organoaluminum decomposition is a method for directly preparing AlH$_3$ without the need for ether solvents [42]. The presence of surfactants enables the obtainment of AlH$_3$ through the decomposition of triethylaluminum under a hydrogen pressure of 10 MPa. The following illustrates the decomposition pathways of Et$_3$Al: Et$_3$Al → Et$_2$AlH → AlH$_3$ → Al. In order to inhibit the further decomposition of generated AlH$_3$ into Al, the second step of decomposition reaction, i.e., Et$_2$AlH → AlH$_3$, needs to be carried out at low temperature. In addition, Tetroctyl ammonium bromide (TOAB) and tetrabylammonium bromide (TBAB) are used as surfactants to stabilize Al/AlH$_3$ particles.

Figure 1 shows the TEM and EDS of as-prepared organo-AlH$_3$ (TOAB and TBAB as the surfactants). When TBAB is used as a surfactant, the particles obtained are relatively large, exceeding 100 nm in size and exhibit a regular spherical shape. In contrast, AlH$_3$ with an irregular morphology and with sizes ranging from 1 to 30 nm can be observed around the periphery of MgH$_2$ particles when TOAB is employed as the surfactant.

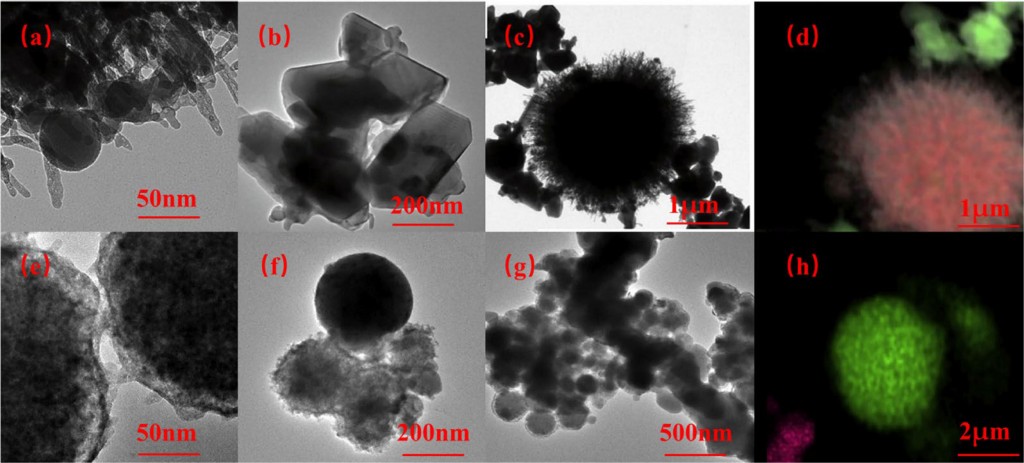

**Figure 1.** TEM and EDS images of AlH$_3$ with TOTB (**a**–**d**) and TBAB (**e**–**h**) synthesized via Et$_3$Al thermal decomposition. Reproduced with permission from Ref. [42]. Copyright 2018, Elsevier.

It is challenging to utilize organoaluminum decomposition to achieve high-purity AlH$_3$ products because of the presence of surfactants. Particularly, when TBAB is used as a surfactant, the synthesized particles exhibit poor stability and low decomposition temperature, limiting its application to propellants.

### 2.3. Other Emerging Methods

#### 2.3.1. Supercritical Synthesis Method

Supercritical synthesis involves the dissolution and reaction of raw materials in supercritical media. Take $AlH_3$ as an example: it can be prepared by reacting activated aluminum with hydrogen in a supercritical liquid at 60 °C for 1 h. [43,44]. The supercritical medium consists of supercritical liquid $CO_2$ in combination with an ether co-solvent. However, due to the use of liquid $CO_2$, the reaction temperature cannot be further increased, and hence limits the reaction activity. The synthetic method is currently in the laboratory stage and still has a long way to go before achieving industrial-scale production.

#### 2.3.2. Organoaluminum Decomposition Method

In the electrolyte solution, active aluminum powder and ionic state hydrogen encounter to trigger an electrochemical reaction, generating $AlH_3$ in the form of precipitation at the bottom of the electrolyte solution.

Zidanet et al. [45] used the Al as the anode and Pt as cathode, respectively, and the whole electrochemical process was realized in the $NaAlH_4$-THF electrolyte. As shown in the reaction paths (Reactions (5) and (6), $AlH_3$ can be produced via two distinct reaction mechanisms at the Al electrode, involving the oxidation of ions and the reaction of $AlH_4^-$ with the Al anode. Based on this, a cycle using electrolysis and catalytic hydrogenation to treat waste aluminum was proposed [46] (see in Figure 2), in which the problem of the high hydrogen pressure and the formation of stable by-products such as LiCl could be avoided. It is noteworthy that the addition of LiCl is notable for its ability to enhance the yield of $AlH_3$; meanwhile, alkali hydride (e.g., LiH, NaH and KH) could be reformed through electrochemical potential.

$$AlH_4^- \rightarrow AlH_3 \cdot nTHF + \frac{1}{2}H_2 \uparrow +e \tag{5}$$

$$3AlH_4^- + Al \rightarrow 4AlH_3 \cdot nTHF + 3e \tag{6}$$

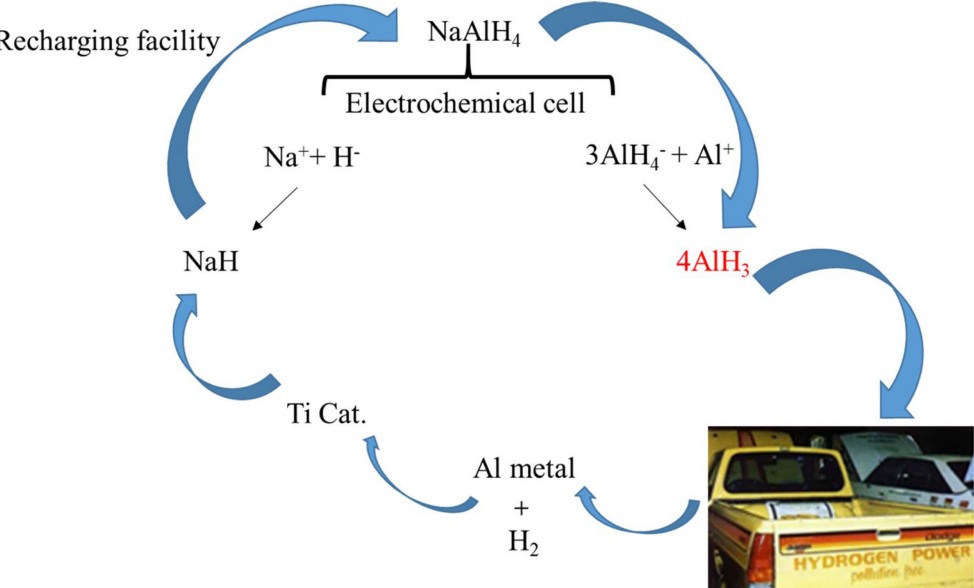

**Figure 2.** Proposed reversible fuel cycle for $AlH_3$.

#### 2.3.3. High Pressure Hydrogenation Method

The absorption of hydrogen by aluminum occurs under high pressure without modification, and could be utilized to prepare $AlH_3$. However, the process necessitates a hydrogen pressure exceeding 700 MPa at ambient temperature. Saitoh et al. [47] studied

the reaction process of aluminum and hydrogen at 10 GPa and 600 °C using in situ X-ray diffraction. As shown in Figure 3, The yellow and red particles represent Al and AlH$_3$ particles, respectively. After 24 h growth, the grain size of AlH$_3$ was about 20 μm. The author also found that the growth process of an AlH$_3$ single crystal will go through three stages: self-pulverization of aluminum (with the increase of reaction time, it is found that the size of aluminum Bragg lattice decreases), hydrogenation of pulverized aluminum (X-ray diffraction image shows a new Bragg lattice), and solid-state grain growth of AlH$_3$.

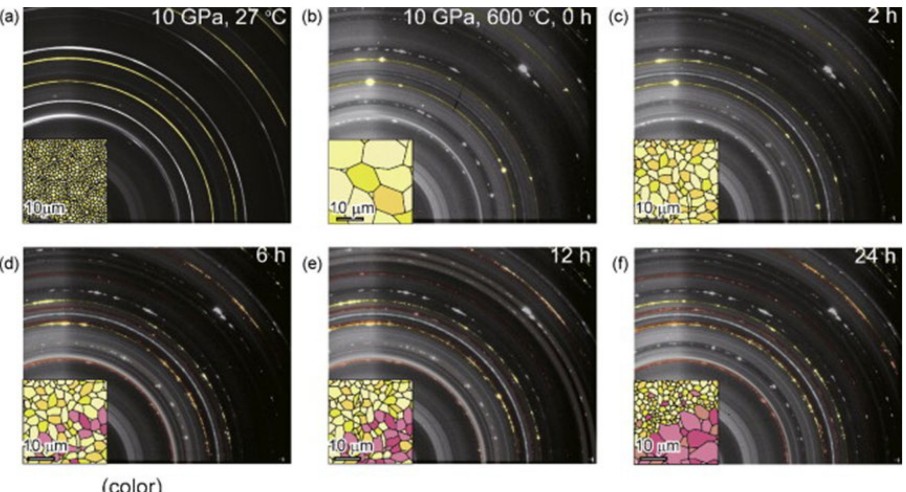

(color)

**Figure 3.** X-ray diffraction patterns and crystal growth process of Al/AlH$_3$ during high-pressure hydrogenation. (**a**) Complete Debye rings from aluminum at 10 GPa, 27 °C; (**b**) Bragg spots from aggregated aluminum (colored in yellow). This image was captured immediately after the sample was heated to 600 °C at 10 GPa; (**c–f**) Diffraction images taken after 2 h, 6 h, 12 h, and 24 h of heat treatment at 10 GPa and 600 °C. Reprinted with permission from Ref. Reproduced with permission from Ref. [47]. Copyright 2010, Elsevier.

The formation of AlH$_3$ through Al hydrogenation is elucidated, wherein AlH$_3$ particles grow into a single crystal through solid-state grain growth. In addition, even at elevated pressures and temperatures, the hydrogenation reaction can be restricted by the chemically stable oxide layer on the surface of Al. The thickness of the surface oxide layer is crucial in altering the growth of single crystals. Given the demanding reaction conditions and low yield, the direct reaction of aluminum and hydrogen under high pressure to produce AlH3 is not deemed a viable option.

### 2.4. Comparison of Synthetic Methods and Its Development Suggestion

To summarize, each method has its own advantages and disadvantages. Table 3 summarizes the common synthesis methods of aluminum hydride and some suggestions for development.

**Table 3.** Comparison of AlH$_3$ synthetic methods and its development proposal.

| Methods | Advantages | Disadvantages | Development Suggestion |
|---|---|---|---|
| Wet chemical methods | 1. High product quality and yield. 2. Simple process. | 1. Large amount of organic solvents used. 2. Flammable raw materials, intermediates and solvents. | Improving the safety control level and synthesis process. |
| Dry synthesis methods | No or small amount of solvent used. | 1. Serious aggregation of AlH$_3$ crystals. 2. Difficult to separate AlH$_3$ from by-products. | Developing suitable reaction conditions, synthesis devices, and purification technology. |

**Table 3.** *Cont.*

| Methods | Advantages | Disadvantages | Development Suggestion |
|---|---|---|---|
| Supercritical synthesis method | Direct reaction of activated Al with H$_2$. | The reaction activity is limited by the relatively low temperature | Break the limits of the reaction temperature by suitable medium. |
| Organoaluminum decomposition method | Cheap raw materials. | 1. Low yield. <br> 2. De-etherification and crystal transformation process of intermediate involved. | Developing low-cost exploratory research. |
| High pressure hydrogenation method | Direct reaction of Al with H$_2$. | 1. Harch reaction conditions. <br> 2. Impurities. <br> 3. Low yield. | Developing mild reaction conditions. |

## 3. Thermodynamics and Kinetics of AlH$_3$

### 3.1. Thermodynamics

Aluminum hydride is known to degrade over time (after three days). This degradation can impact the stability and effectiveness of AlH$_3$ as a hydrogen storage material. α-AlH$_3$ can decomposes directly into Al and H$_2$ through a single endothermic step (7). Sinke et al. [48] first investigated the thermodynamic properties of α-AlH$_3$, and the formation enthalpy was measured to be −11.4 kJ mol$^{-1}$ at 25 °C. The less stable polymorphs α', β, and γ-AlH$_3$ will undergo an exothermic transition to α-AlH$_3$ (8) at around 100 °C [49–51], whereas δ, ε, and ζ-AlH$_3$ would direct decomposition into Al without undergoing any crystal transformation. However, the direct decomposition of α'-AlH$_3$ and γ-AlH$_3$ into Al and H$_2$ was also observed (9) [50,52–54]. Liu et al. [53] studied the decomposition mechanisms of γ-AlH$_3$ and suggested that the outer layer of the γ-AlH$_3$ particle is more likely to undergo direct decomposition, while the inner part tends to become the more stable α-AlH$_3$ before decomposition (Figure 4). The enthalpies of the transition from α', β, and γ-AlH$_3$ to α-AlH$_3$ were measured to be 1.6, 1.5, and 2.8 kJ/mol, respectively [50,55]. The formation enthalpy of approximately −10 kJ/mol was estimated for α-AlH$_3$ [55], and the value ranges from −9.5 to −9.9 kJ/mol for β or γ-AlH$_3$, while a low formation enthalpy of −2.5 kJ/mol was reported for α'-AlH$_3$ [50].

$$\alpha - AlH_3 \rightarrow Al + 3/2H_2 \uparrow \tag{7}$$

$$\beta - AlH_3/\gamma - AlH_3/\alpha' - AlH_3 \rightarrow \alpha - AlH_3 \tag{8}$$

$$\gamma - AlH_3/\alpha' - AlH_3 \rightarrow Al + 3/2H_2 \uparrow \tag{9}$$

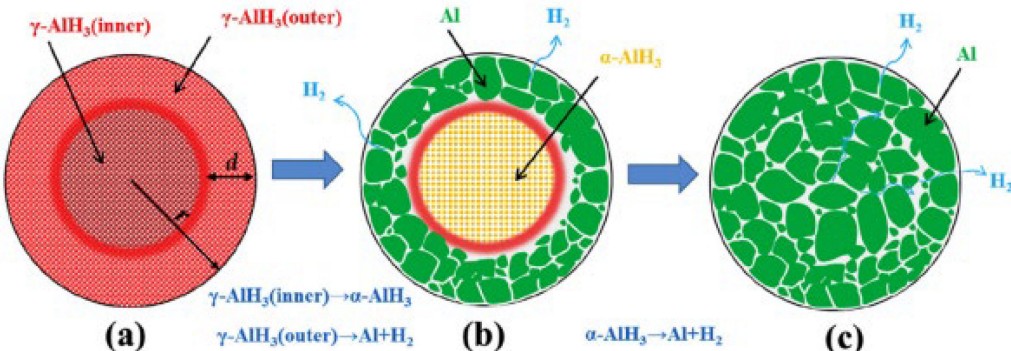

**Figure 4.** The dehydrogenation mechanism of γ-AlH$_3$. (**a**) as-synthesized γ-AlH$_3$; (**b**) after partial decomposition; (**c**) after full decomposition. reproduced with permission [53]. Copyright 2017, Elsevier.

Bulk AlH$_3$ is thermodynamically unstable (metastable) and is only kinetically stable under ambient conditions. Nanoscaling is a well-known strategy for destabilizing bulk metal hydrides, yielding faster rates of H$_2$ desorption, reduced formation of parasitic

intermediates, and greater reversibility. Stavila et al. [56] developed a new strategy for thermodynamic stabilization of metal hydrides by incorporating alane clusters within the pores of covalent triazine frameworks (CTF). The highly unfavorable thermodynamics of direct aluminum hydrogenation can be overcome by stabilizing alane within a nanoporous bipyridine-functionalized covalent triazine framework (AlH$_3$@CTF-bipyridine). The material desorb hydrogen at temperatures as low as 95 °C, with ca. 2/3 of the total hydrogen released within less than 10 min from the onset of dehydrogenation and can be partially rehydrogenated at 70 MPa (700 bar) H$_2$ (Figure 5). The results demonstrate that the strategy of thermodynamic stabilization through concomitant nanoconfinement could enable reversibility in high-capacity metastable metal hydrides.

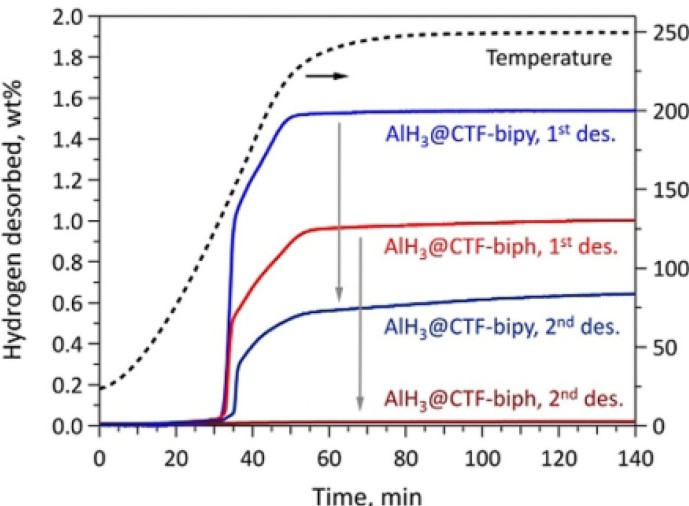

**Figure 5.** Sieverts data for AlH$_3$@CTF-biph and AlH$_3$@CTF-bipy samples prior and after the 700 bar (70 MPa) H$_2$ absorption cycle. Reproduced with permission [56] Copyright 2021, Wiley Online Library.

### 3.2. Kinetics

Herley et al. [57–59] firstly measured the thermal decomposition behavior of AlH$_3$ provided by DOW Chemical Company. The isothermal decomposition and TG-DSC curves of AlH$_3$ are shown in Figures 6a and 6b, respectively. The decomposition process is characterized by an endothermic step, and the isothermal decomposition curves of AlH$_3$ are usually divided into three parts: induction period, acceleratory period, and decay period. The induction period corresponds to the beginning of decomposition, and could be attributed to the crack of the surface layer. The acceleratory period results from rapid hydrogen release due to the multi-dimensional growth of aluminum [27]. The decay period marks the completion of the decomposition process. [23,27]. Therefore, through managing the time of induction period, the decomposition rate can be controlled [60]. In essence, controlling the formation of Al nuclei is the crucial factor in altering the decomposition rate.

The thermal reaction of AlH$_3$ in an air atmosphere is complex and can be categorized into three stages: dehydrogenation and passivation, primary oxidation, and secondary oxidation. It is noteworthy that the passivation process involving Al → Al$_2$O$_3$, occurs almost simultaneously with the dehydrogenation reactions. An amorphous Al$_2$O$_3$ layer formed on the particles' surface by passivation encapsulates the contained hydrogen, and leads to incomplete dehydrogenation.

Milekhin et al. [61] employed TG-DSC, SEM, and EDS to study the dehydrogenation and oxidation kinetics and mechanisms of micron-sized $\alpha$-AlH$_3$ under various atmospheres, including nitrogen, argon, and air. The results indicated that the samples were stable below 150 °C, but can be decomposed between 150 and 180 °C. Figure 6 illustrates the theoretical kinetic model of AlH$_3$ in an oxidative environment.

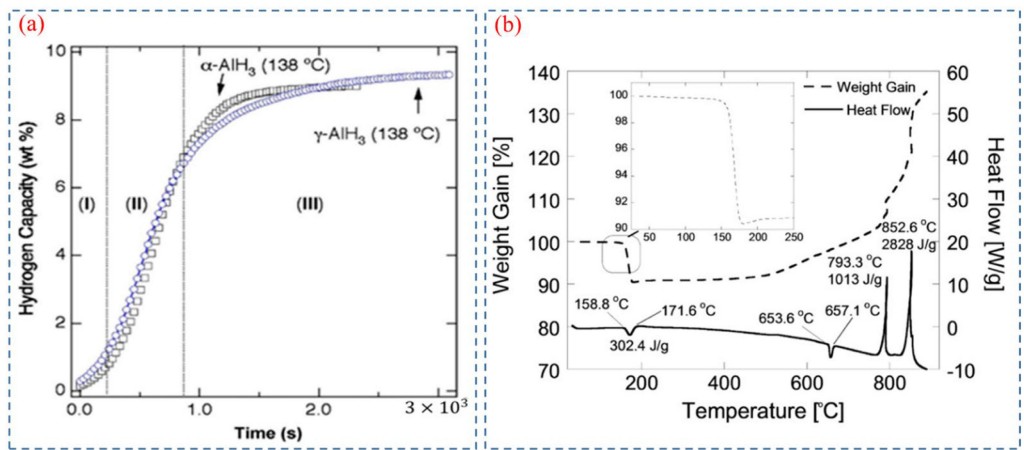

**Figure 6.** (**a**) Isothermal decomposition curves of α-AlH₃ and γ-AlH₃ [51]. Copyright 2007, Elsevier. (**b**) TGA and DSC of neat AlH₃ in an argon atmosphere [1]. Copyright 2019, Wiley Online Library.

Figure 7a depicts a schematic diagram illustrating the hydrogen decomposition on the surface of Aluminum hydride [62]. The process includes breakage of the surface oxide layer (e.g., Al₂O₃), recombination of hydrogen, and growth of aluminum [11]. The oxide layer surrounding AlH₃ particles is prone to cracking at elevated temperatures, as the volumetric thermal expansion of AlH₃ is approximately double that of amorphous and crystalline Al₂O₃. The results also indicate that hydrogen desorption begins only when the surface oxide layer breaks and the free AlH₃ on the surface of particles contact with the gas phase.

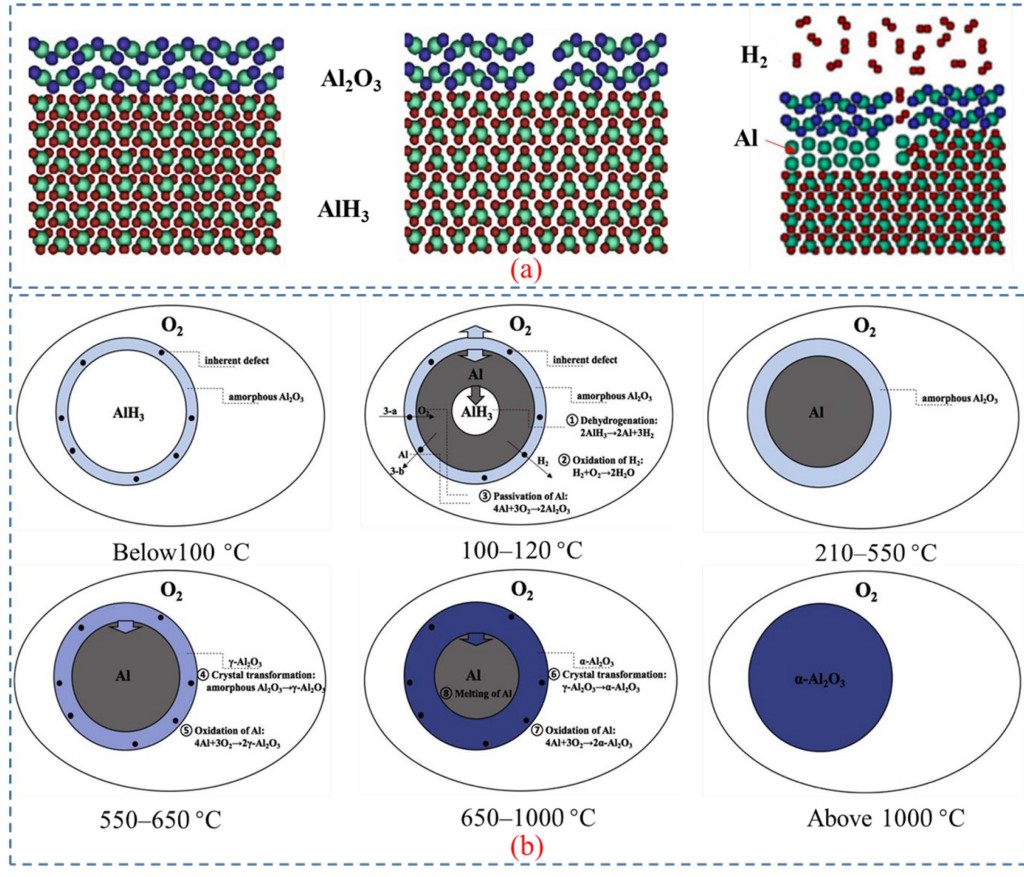

**Figure 7.** (**a**) The mechanism underlying hydrogen release from AlH₃ [62]. Copyright 2010, AIP Publishing. (**b**) The mechanism of dehydrogenation and oxidation of AlH₃ at different temperatures [10]. Copyright 2020, Elsevier.

Liu et al. [63] investigated the thermal oxidation of $AlH_3$ and proposed dehydrogenation/passivation mechanism. $AlH_3$ usually has a protective layer of $Al_2O_3$ on its surface, which inhibits the dehydrogenation reaction of $AlH_3$ at room temperature [44]. With the increase of temperature, cracks are formed in the $Al_2O_3$ layer, exposing the $AlH_3$ surface and causing the occurrence of incomplete dehydrogenation. Since the distance between Al atoms in $AlH_3$ is shorter than that in elemental Al, significant volume shrinkage will occur during the conversion of $AlH_3$ to Al [11,64], resulting in the formation of pores on the particle surface and further increase the exposed $AlH_3$.

The transformation process of Al-containing substances during dehydrogenation and oxidation at different temperatures is shown in Figure 7b. The transition follows the route: $\alpha$-$AlH_3 \rightarrow$ Al $\rightarrow \gamma$-$Al_2O_3 \rightarrow \alpha$-$Al_2O_3$. The reaction consists of decomposing of $AlH_3$, oxidizing of $H_2$, passivating of Al, and the crystal transition of $Al_2O_3$ to $\gamma$-$Al_2O_3$ and $\gamma$-$Al_2O_3$ to $\alpha$-$Al_2O_3$.

Dehydrogenation and oxidation of $AlH_3$ are influenced by three factors: the oxide layer, the surrounding atmosphere, and temperature. As mentioned above, decomposition only takes place when the oxide layer fractures, allowing free $AlH_3$ to come into contact with the surrounding atmosphere. Nakagawa et al. [65] investigated the effect of the thickness of the oxide layer on the dehydrogenation kinetics of $\alpha$-$AlH_3$. The TEM results (Figure 8) indicated that the thickness of $Al_2O_3$ seems to increase with exposure time through diffusion-controlled growth. Moreover, the peak temperatures during hand milling (135 °C) are significantly lower than those without milling (150 °C) under an argon atmosphere. This is believed to be due to the $Al_2O_3$ layer on the surface of $AlH_3$, which may delay the dehydrogenation process. The $Al_2O_3$ and $Al(OH)_3$ layer can also affect the dehydrogenation of $AlH_3$. Yu et al. [66] prepared a novel core–shell structured $\alpha$-$AlH_3@Al_2O_3@Al(OH)_3$ through hydrochloric acid leaching, which does not decompose under environmental conditions for several days. These findings suggest that the presence of an oxide layer can postpone the dehydrogenation process.

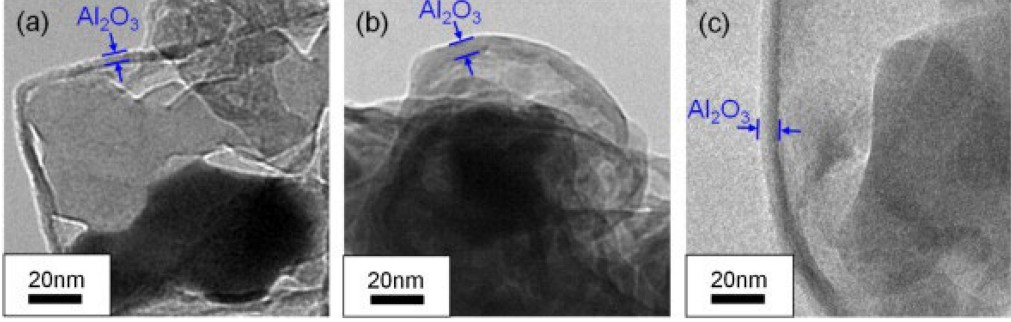

**Figure 8.** The TEM images of $Al_2O_3$ films on $AlH_3$ particles (**a**) without exposure to air, (**b**) one day after exposure to air, (**c**) seven days after exposure to air [65]. Copyright 2013, Elsevier.

The thermal reaction of $AlH_3$ is significantly influenced by the surrounding atmosphere, because different products could be generated with various gaseous, such as $Al_2O_3$ and AlN. Additionally, the dehydrogenation of $AlH_3$ is competitive with Al oxidation in an oxidative atmosphere. As illustrated in Figure 9, the surface agglomeration and roughness of particles in oxidative atmosphere are more obvious under elevating temperature.

The dehydrogenation and oxidation properties of $AlH_3$ are also affected by the heating rate and temperature. The impact of the heating rate was examined using a non-isothermal method. The results indicated that both the initial and final temperatures, as well as the final mass loss of the hydrothermal solution, decreased as the heating rate decreased This reason can be attributed to the slower hydrogen diffusion through particles or nucleation at aluminum sites at lower temperatures and heating rates, resulting in some hydrogen remaining trapped in the particles [23].

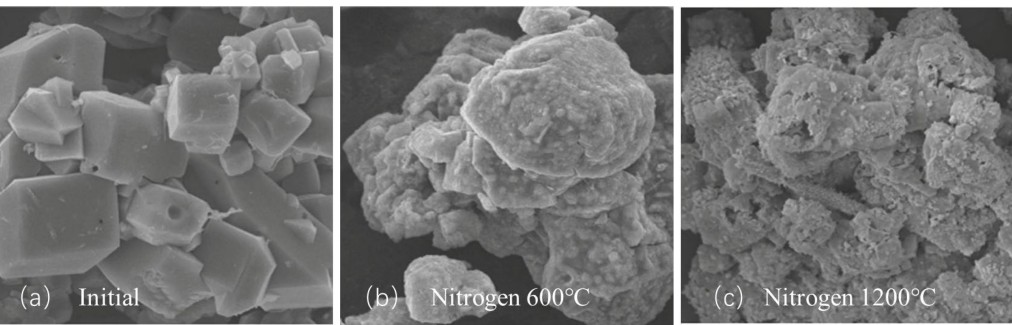

**Figure 9.** Microphotographs of the AlH$_3$ samples: (**a**) initial sample (**b**,**c**) after heating to 600 °C and 1200 °C in nitrogen medium, respectively. Reproduced with permission [61]. Copyright 2015, Springer.

## 4. Physical and Chemical Modification of AlH$_3$

Due to the low thermal stability of AlH$_3$ in ambient atmosphere, dehydrogenation will occur easily. The factors contributing to the instability of AlH$_3$ can be categorized into intrinsic and extrinsic causes. Regarding intrinsic causes, AlH$_3$ consists of two highly reducing elements, aluminum and hydrogen, with a small enthalpy of formation and positive Gibbs free energy [26,67,68]. Therefore, AlH$_3$ is in metastable state, and has a spontaneous tendency to decompose into aluminum and hydrogen from thermodynamic perspective.

On the other hand, impurities like LiCl, NaBH$_4$, LiBH$_4$, and LiAlH$_4$, as well as unstable polymorphs formed during AlH$_3$ synthesis can also reduce its thermal stability [69]. Apart from this, other extrinsic factors, e.g., humidity, light, oxygen also make AlH$_3$ unstable.

When AlH$_3$ is used as a high-energy additive in propellants, to ensure its stable storage without decomposition and reduce the potential hazards during storage, transportation and use, it is necessary to stabilize AlH$_3$, for which multiple methods are available, including surface passivation, doping, surface coating.

### 4.1. Surface Passivation Methods

In this method, some solutions, e.g., aqueous organic solution, buffer solution, and various acid solutions able to react with $\alpha$-AlH$_3$, are generally used. For one thing, the solutions can remove the synthetic impurities and unstable crystal forms. For another, they can shield against external stimuli by developing passivation layers of Al$_2$O$_3$ or Al(OH)$_3$ on the surface of $\alpha$-AlH$_3$.

Nile et al. [70] treated $\alpha$-AlH$_3$ with n-butylamine containing a small amount of water. It took 60 days for the decomposition rate to reach 1%, while the untreated $\alpha$-AlH$_3$ reached the same decomposition rate in 13.5 days, indicating the stabilization effect of the treatment.

Robert et al. [71] used neutral buffer solution KH$_2$PO$_4$ and NaOH to treat $\alpha$-AlH$_3$ at 70 °C. The contents of impurities, e.g., C, Cl, and Li, in the particles were significantly reduced after 15 min, and the decomposition rate of $\alpha$-AlH$_3$ was also decreased, it took 26 days for the decomposition rate to reach 1% compared with 8 days without treated.

The $\alpha$-AlH$_3$ particles can also be stabilized using acid solutions. Typically, the acid choices include hydrochloric acid, hydrogen fluoride acid, hydrogen bromide acid, and others, with hydrochloric acid being the most preferable. Petrie et al. [17] used hydrochloric acid solution to clean $\alpha$-AlH$_3$. Meanwhile, the pickling process can also form Al$_2$O$_3$ or Al(OH)$_3$ layer on the surface of the $\alpha$-AlH$_3$, which plays a stabilizing role. It took 12.7 days for $\alpha$-AlH$_3$ treated with hydrochloric acid solution to reach 1% decomposition at 60 °C, while the untreated $\alpha$-AlH$_3$ only takes 9.3 days. Figure 10 shows the SEM images of $\alpha$-AlH$_3$ before and after hydrochloric acid treatment. The results demonstrate that acid etching can eliminate cracks and impurities, expected to be beneficial to thermal stability of AlH$_3$ products [24].

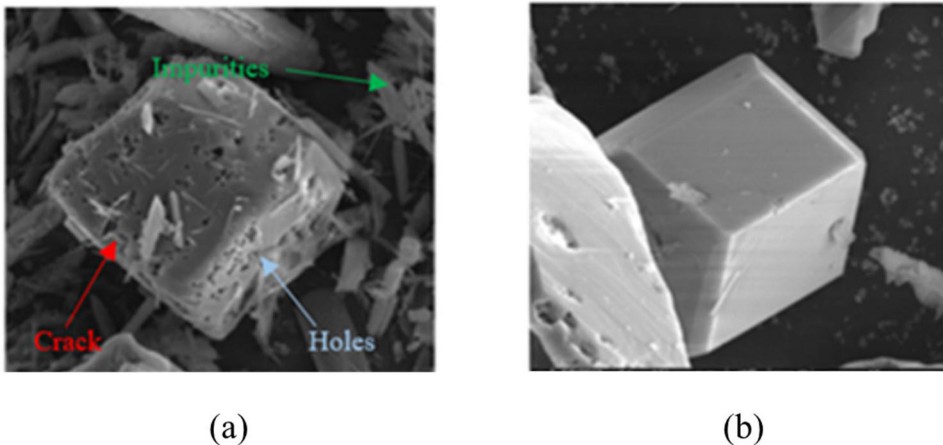

**Figure 10.** SEM images of α-AlH$_3$ (**a**) before and (**b**) after hydrochloric acid treatment.

The mechanism of hydrochloric acid stabilization of α-AlH$_3$ was investigated by Yu et al. [66]. They found that hydrochloric acid plays an important role in accelerating the hydrolysis reaction of AlH$_3$ to generate honeycomb-like structures which can generate integrity and dense oxide layers. Moreover, the detailed acid stabilization mechanism could be divided into surface oxidation, oxide layer rupture, hydrolysis reaction, and transition of oxide layers, as depicted in Figure 11.

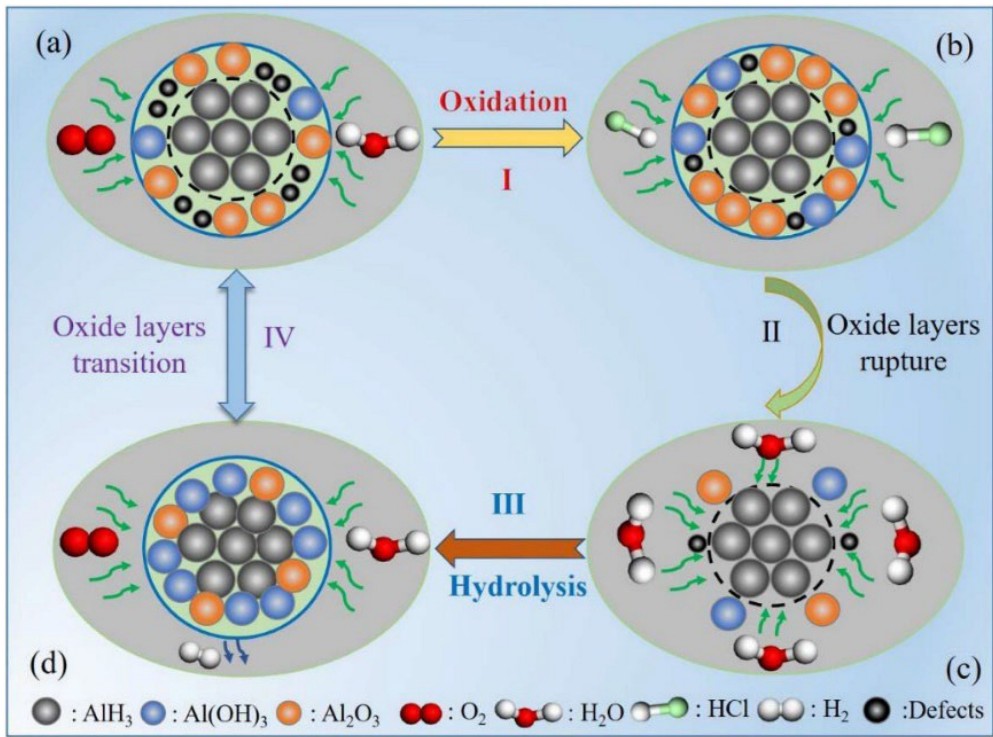

**Figure 11.** The stabilization mechanisms by acid passivation for AlH$_3$ [66]. (**a**) original structure; (**b**) surface oxidation; (**c**) oxide layers rupture and (**d**) honeycomb-like structure. Copyright 2022, Elsevier.

Additionally, surface passivation of AlH$_3$ can be achieved by heating it in air or an oxidizing atmosphere [72]. Despite the benefits to improve the thermal stability, the presence of an oxide layer by various passivation methods will inevitably cause the energy loss of α-AlH$_3$ and affect the combustion performance.

## 4.2. Doping Methods

The thermal stability of AlH$_3$ can be improved by doping following two approaches. One approach aims to eliminate the highly active sites in α-AlH$_3$ crystal structure, hence increasing the activation energy of decomposition reaction. Norman et al. [73] added magnesium powder to the reaction of preparing α-AlH$_3$, and the X-ray diffraction results showed varying degrees of expansion for the crystal structure of α-AlH$_3$. It took 26 days for α-AlH$_3$ doped with 2% magnesium powder to reach 1% decomposition rate in an anhydrous nitrogen atmosphere at 60 °C, while the undoped one only took 5 days. Cianciolo et al. [74] prepared α-AlH$_3$ particles by doping α-AlH$_3$ ether complexes with Hg. The experimental results indicated that the decomposition rate of α-AlH$_3$ particles doped with 0.02% Hg is 0.4% at 100 °C for 24 h, far lower than the value of 9.4% of undoped particles under the same conditions. Furthermore, doping with other metal elements, such as Si, can also enhance the thermal stability of AlH$_3$ [74].

In theory, the decomposition of AlH$_3$ generates an electron hole caused by the formation of positively charged ions, while the Al$^{3+}$ ions can further catalyze the surface decomposition of the bulk sample [75]. Therefore, the other approach is inhibiting the decomposition process by increasing the Lewis acid, Lewis base, or other compounds that can coordinate with Al$^{3+}$ ions. It has been reported that the addition of such free radical inhibitors can enhance the thermal stability of AlH$_3$ by a factor of 10 to 20 [2]. For example, Alan et al. [76] doped of AlH$_3$ using phenothiazine (PTA) or Mercaptobenzothiazole (MBT, as shown in Figure 12), making its decomposition rate only 0.97% after 27 days of storage at a constant temperature of 60 °C, far lower than that of untreated samples. Roberts et al. [77] doped aryl or alkyl-substituted silanols in the preparation process of α-AlH$_3$. From their results, 8 and 4 days were reported for the decomposition rate to reach 1% for the doped and undoped α-AlH$_3$, respectively.

**Figure 12.** Stabilization mechanism of Mercaptobenzothiazole (MBT).

Similar to surface passivation methods, doping of α-AlH$_3$ also results in a loss of energy due to decrease in the crystal purity of α-AlH$_3$.

## 4.3. Surface Coating Methods

The surface coating method has received wide attention in recent years because it does not destroy the chemical reactivity of α-AlH$_3$ [70,77]. Typically, the surface of AlH$_3$ is coated with inert and hydrophobic materials to create a core–shell structure, thereby altering its surface properties [78]. The structure not only prevents direct contact between the inner AlH$_3$ and humidity or oxygen but also reduces the number of active sites on the surface, thereby increasing the activation energy of the decomposition reaction [79]. Currently, coating materials mainly consist of inorganic and organic molecules, metal oxides, organic polymers, carbon materials, and some energetic components.

### 4.3.1. Inorganic and Organic Molecules

Norman et al. [80] used small gaseous (such as $N_2F_4$, NO) and liquid molecules (such as $Al_2S_3$, $AlCl_3$) to form a coating layer on the surface of $\alpha$-$AlH_3$ through an adsorption process. As a typical example, under 100 °C, the decomposition amount of the coated sample with NO molecule is reduced from 5.63% to 0.21%, for a duration of 7 h. In addition, after treatment with these small molecule stabilizers, $\alpha$-$AlH_3$ shows good compatibility with other propellant components, such as ammonium perchlorate and nitrocellulose.

Schmidt et al. [18] used organic compounds containing nitrile groups to coat $\alpha$-$AlH_3$, and the stability of $AlH_3$ was found to be enhanced through the bond interaction of nitrile groups and free radicals on its surface. Besides, the compatibility between $\alpha$-$AlH_3$ and propellant components was also improved.

Qin et al. [14] coated $\alpha$-$AlH_3$ with stearic acid, and no ignition was observed for the coated sample even when the value of electrostatic sensitivity($E_{50}$) reached the test limit of 5390 mJ. In comparison, the $E_{50}$ of $\alpha$-$AlH_3$ without coating is only 367 mJ.

Shi et al. [81] used a silane coupling agent, A171, to modify the surface of $\alpha$-$AlH_3$. The surface was uniformly covered with a thin organic layer via the coating process shown in Figure 13. After coating, the standard 100 °C thermostatic thermal stability and activation energy of $\alpha$-$AlH_3$ increased up to 95.25 min and 4.32 kJ mol$^{-1}$, respectively.

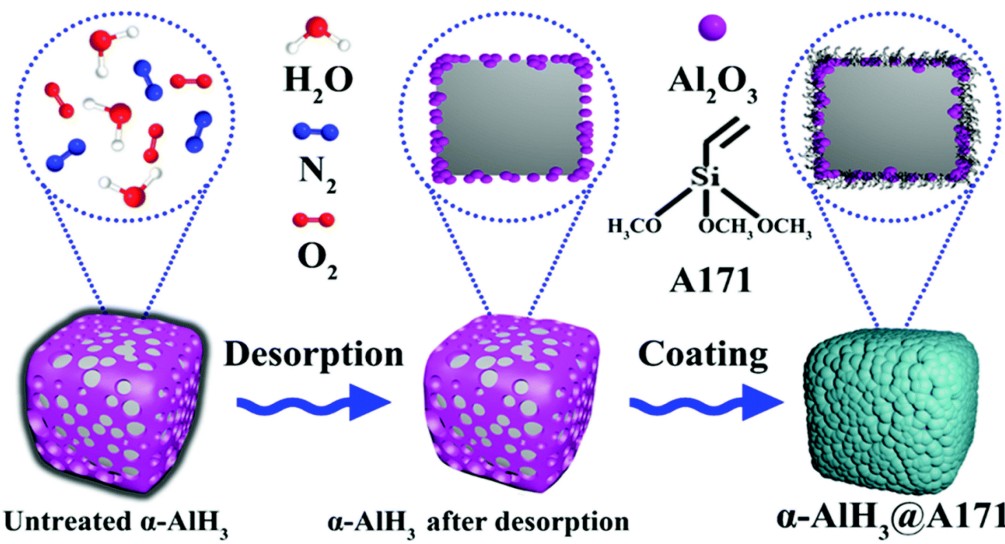

**Figure 13.** The preparation process of the $\alpha$-$AlH_3$@A171 composite [81]. Copyright 2022, Royal Society Chemistry.

### 4.3.2. Metal Oxides

Kempa et al. [82] found that the surface aluminum oxide layer of $\alpha$-$AlH_3$ can effectively isolate water and oxygen molecules in the environment so that the crystal structure of $\alpha$-$AlH_3$ does not change significantly for a long time. In recent years, researchers have tried to control the coating structure by atomic layer deposition (ALD) and other advanced techniques to achieve accurate control of the thermal stability of $AlH_3$ [83].

Chen et al. [12] used ALD to deposit a layer of amorphous $Al_2O_3$ on the surface of $\alpha$-$AlH_3$. The results showed that with the increase of coating cycles from 100 to 200, the decomposition temperature increased from 158.1 °C to 161.2 °C compared to 153.4 °C for the uncoated sample. Figure 14 displays the SEM images of $\alpha$-$AlH_3$@$Al_2O_3$ particles after the hydrothermal aging test. It can be clearly seen that the surface of $\alpha$-$AlH_3$ particles after 200 coating cycles becomes much smoother than the uncoated one. Moreover, its friction sensitivity decreases from 96% to 68% after the hydrothermal aging test. The $Al_2O_3$ films can effectively impede the transfer of frictional heat to the $AlH_3$ core, thereby reducing the friction sensitivity of the core–shell structure.

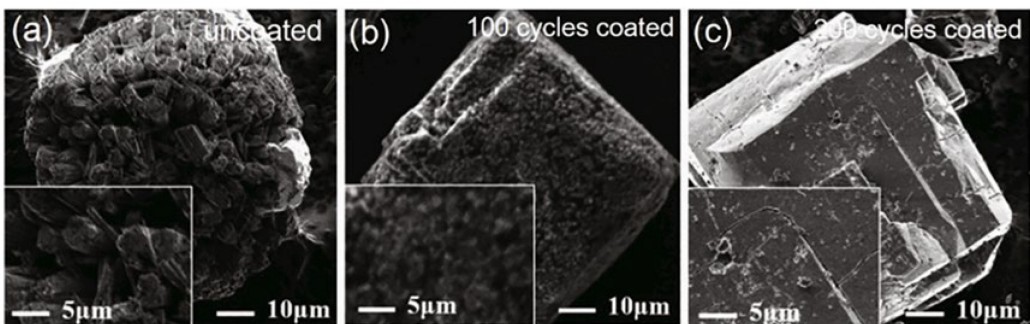

**Figure 14.** SEM images of $\alpha$-AlH$_3$ stabilized by ALD method before and after. Hydrothermal aging test after ALD method (**a**) 0 cycles; (**b**,**c**) 100 and 200 cycles, respectively [12]. Copyright 2017, AVS.

### 4.3.3. Organic Polymers

Due to the non-energetic properties of small molecules as a coating layer for stabilization of AlH$_3$, the overall energy performance of the propellant will be attenuated. Therefore, the coating layer is preferably selected from the energetic components, such as nitrocellulose (NC). Flynn et al. [20] applied $\alpha$-AlH$_3$ particles coated with nitrocellulose to a double-base solid propellant. As a result, the decomposition rate of AlH$_3$ was only 0.63% at 50 °C for 90 days, indicating that it had a significant stabilization effect.

Cai et al. [16] used supercritical fluid technology to cover the surface of $\alpha$-AlH$_3$ with fluoroelastomer (FE26) uniformly. Total formation enthalpy ($\Delta H_f$) of alane and alane/FE26 increased from $-15.7$ kJ/mol to $-21.0$ kJ/mol at heating rate of 20 °C, while Gibbs free energy ($\Delta G_f$) decreased from 44 kJ/mol to 38.7 kJ/mol, indicating increased thermal stability. In electric sparkle sensitivity test, the E$_{50}$ value of alane is 63.71 mJ, and the value increased up to 85.24 mJ after 5% FE26 is coated on alane. This result indicates that after coating the FE26, safe electrostatic discharge can be achieved.

### 4.3.4. Carbon Materials

Su et al. [84] reported a novel method of coating AlH$_3$ with carbon nanotubes (CNT), where re-nucleation and growth of $\alpha$-AlH$_3$ on the surface of the coating material, rather than the formation of the coating material on the surface of $\alpha$-AlH$_3$ particles, occurred.

Xing et al. [15] attempted to improve the thermal stability of $\alpha$-AlH$_3$ by using fullerene stabilizers (C$_{60}$), which allows $\alpha$-AlH$_3$ to be stored at 60 °C for 3 months with a decomposition rate of less than 1%.

Li et al. [85] used the solvent-antisolvent method to coat $\alpha$-AlH$_3$ with graphene oxide (GO). The $\alpha$-AlH$_3$-GO material obtained was successfully applied to the solid propellant slurry, for which the mechanical impact sensitivity of slurry was effectively reduced, evidenced by an increase of the I$_{50}$ value from 7.3 J to 12.1 J.

### 4.3.5. Energetic Components

As mentioned above, increasing the content of this materials would undoubtedly reduce the energy density and detonation property of the propellants. Therefore, in order to achieve higher energy density, the coatings layer can be selected from stable energetic components.

Yu et al. [86] prepared the homogenous composites of AlH$_3$ and commonly used oxidizer (HMX, CL-20, and AP) using the in situ recrystallization method, for which thermal stability, compatibility, and ignition performance were investigated. The results showed that the initial decomposition temperatures of AlH$_3$/oxidizer are increased by about 16 °C, and the induction time of the dehydrogenation of AlH$_3$ is extended by 1.2 times. Such a stabilization effect of the oxidizers can be attributed to strong hydrogen bonding. Furthermore, the flame radiation intensity of each composite was enhanced, and the most intensive value of AlH$_3$/CL-20 is 3.4 times that of pure AlH$_3$. Moreover, the

crucible is even broken (marked with a white circle) when using AlH$_3$/AP composites as trigger for water-cooled CO$_2$ laser ignitor, as depicted in Figure 15.

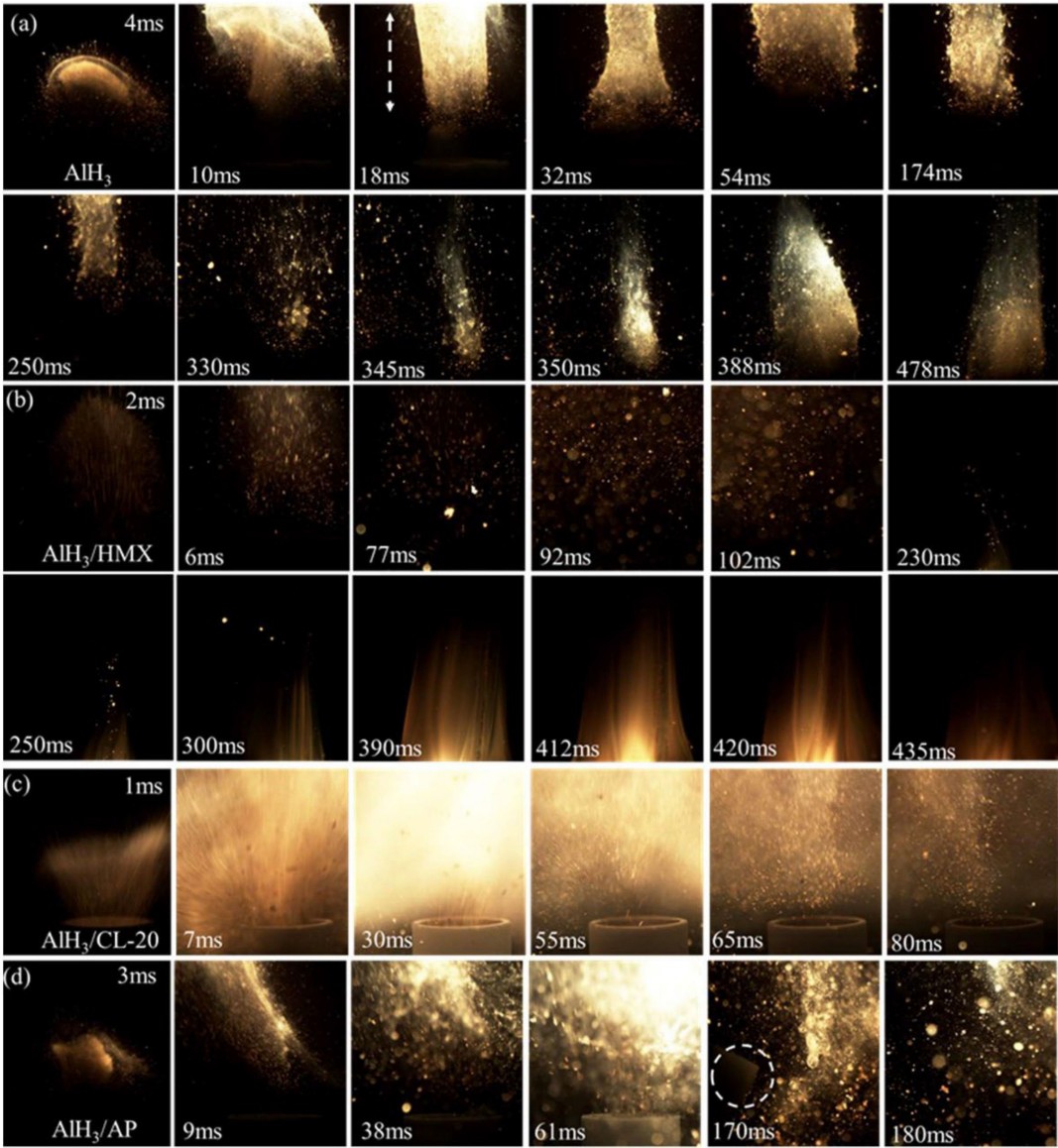

**Figure 15.** Flame images of raw AlH$_3$ and AlH$_3$/oxidizer composites: (**a**) AlH$_3$, (**b**) AlH$_3$/HMX, (**c**) AlH$_3$/CL-20, and (**d**) AlH$_3$/AP [86]. Copyright 2023, American Chemical Society.

Yu et al. [87] prepared fluoropolymer- and AP-coated AlH$_3$-based composites (AHFPs) via a spray-drying technology aiming to improve the stability and combustion performance of AlH$_3$. The results showed that the initial decomposition temperatures of AHFPs were increased by 17 °C, compared with pure AlH$_3$. Moreover, the decomposition induction time was increased by 1.82 times compared to raw AlH$_3$, indicating improved stability by the coatings of PFPE and AP. The maximum flame radiation intensity of AHFPs-30% is 7.71 times that of pure AlH$_3$, suggests that the combustion performance of AlH$_3$ has also been enhanced. Table 4 summarizes the primary reported stabilization methods and properties of stabilized AlH$_3$.

At present, the research of stabilizing of AlH$_3$ mainly focus on the surface coating method, involve the screening of the coating agent, especially energetic materials. However, how to accurately control the coating thickness so as to have better compatibility with the propellant components is the key for further research.

**Table 4.** The methods used for stabilizing and the resulting properties of $AlH_3$.

| Methods | Materials Used for Stabilization | $\alpha$-$AlH_3$ before Stabilization | $\alpha$-$AlH_3$ after Stabilization |
|---|---|---|---|
| Surface passivation | Mg and N-butylamine ($H_2O$) [15,70] | 1% decomposition for 13.5 d at 60 °C | 1% decomposition for 60 d at 60 °C |
| | $C_2H_5OH$ (98%) [70] | - | 0.1% decomposition for 35~40 d at 60 °C |
| | Mg and $KH_2PO_4$/NaOH [77] | - | 0.25% decomposition for 49 d at 60 °C |
| | Mg, N-butylamine ($H_2O$) and $KH_2PO_4$/NaOH [77] | 1% decomposition for 33 d at 60 °C | 1% decomposition for 43 d at 60 °C |
| | HCl solution [17] | 1% decomposition for 9.3 d at 60 °C | 1% decomposition for 12.7 d at 60 °C |
| | air (60 °C for 250 h) [72] | 5% decomposition for 95 min at 115 °C | 5% decomposition for 237 min at 115 °C |
| Doping | MBT/PTA [76] | 7.5% decomposition for 14 d at 60 °C | PTA: 0.97% decomposition 60 °C for 27 d; MBT:0.6% decomposition 60 °C for 17 d |
| | Hg [74] | 9.4% decomposition for 24 h at 100 °C | 0.4% decomposition for 24 h at 100 °C |
| | Mg [73] | 1% decomposition for 5 d at 60 °C | 1% decomposition for 26 d at 60 °C |
| | Si [74] | 1% decomposition for 4 d at 60 °C | 1% decomposition for 8 d at 60 °C |
| Surface coating | NO, $N_2F_4$ [80] | 5.63% decomposition for 7 h at 100 °C | 100 °C for 7 h, 0.21% decompose- tion |
| | $Al_2S_3$ [80] | Rapid decomposition for 2 h at 60 °C after store at room temperature for 114 d | 0.13% decomposition for 22 h at 60 °C after store at room temperature for 114 d |
| | SA [18] | $E_{50}$: 367 mJ | $E_{50}$: 5390 mJ |
| | Diphenylacetylene [18] | 1% decomposition for 13 d | 0.84% decomposition for 48 d |
| | Nitrocellulose [20] | Rapid decomposition for 30 d at 50 °C, or for 10 d at 60 °C | 0.3% decomposition for 90 d at 50 °C; 0.63% decomposition for 90 d at 60 °C |
| | FE26 [16] | $E_{50}$: 63.7 mJ | $E_{50}$: 85.24 mJ |
| | $Al_2O_3$ [12] | 7.8% decomposition for 12 h at 70 °C | 0.49% decomposition for 12 h at 70 °C |
| | GO [85] | $I_{50}$: 7.3 J | $I_{50}$: 12.1 J |
| | $C_{60}$ [15] | - | <1% decomposition for 90 d at 60 °C |

Note: MBT: 2-Mercaptobenzothiazole; PTA: Phenothiazine; $E_{50}$: electrostatic spark sensitivity; $I_{50}$: impact sensitivity; SA: Stearic acid.

## 5. High Energy Fuel for Solid Propellant

Compared with aluminum powder, a wildly metal fuel adopted to improve the energy characteristics of composite propellants [88,89], $AlH_3$ has extraordinary potential in the application of solid propellant. NASA CEA results indicate that replacing aluminum with $AlH_3$ will increase specific impulse by 10% [90] over AP/HTPB rocket propellant and reduce flame temperature by 5% [91]. Furthermore, $AlH_3$ produces more $H_2$ and $H_2O$ in the combustion products, reducing the mole fraction of combustion product. Shark et al., found the use of $AlH_3$ hydride additives could raise the overall specific impulse of DCPD/RGHP by 4% [92]. Maggi et al., demonstrated a 6–8% gain in gravimetric specific impulse gain for the aluminum and lithium aluminum hydrides when aluminized AP/HTPB compositions were chosen as a Reference [91].

The ballistic and physical properties of propellants based on $AlH_3$ were investigated experimentally [3]. As depicted in Figure 16a–c, the flame intensity of $AlH_3$-based propellants is significantly lower compared to micron and nano-aluminized solid propellants, indicating a lower adiabatic temperature. Another difference lies in the aggregation/agglomeration processes on the burning surface. Figure 15d illustrates that the high burning rate of $AlH_3$-based propellants can effectively prevent the aggregation phenomenon.

Bazyn et al. [93] studied the combustion characteristics of $AlH_3$ under the condition of a solid rocket motor. The experimental results show that the decomposition time of $AlH_3$ is exponentially correlated with temperature, which satisfies the Arrhenius type rate equation.

Although $AlH_3$ has been used as energy fuel for solid propellant. The intrinsic metastability affects its further practical application. The easy decomposition of $AlH_3$ causes concern for the safe storage and continuous combustion of the propellant system.

**Figure 16.** (**a**–**c**) The flame structures observed during the combustion of AlH$_3$-based propellants; (**d**) the burning surface [3]. Copyright 2007, Elsevier.

## 6. Conclusions and Suggestions

AlH$_3$ is a binary metal hydride with seven known polymorphs among which $\alpha$ is the most stable. Due to the high volumetric energy density of AlH$_3$, the introduction into solid propellant is helpful to provide a large amount of energy via combustion. In this paper, the synthesis methods, key properties, and tuning of AlH$_3$ are presented, with the following conclusions drawn:

1.  The commonly used methods of synthesizing AlH$_3$ are wet chemical synthesis and mechano-chemical method, both of which are faced with the difficulty in achieving high crystal purity of the product. Adding crystalline inducer helps to control the proportion of crystal types, but the presence of an inducer in the product will lead to a decrease in the thermal stability of AlH$_3$. Therefore, highly effective inducers are easy to separate, and need to be developed.

2.  From a kinetics perspective, the acceleratory period is a critical stage that governs the decomposition rate of aluminum hydride, primarily due to the multi-dimensional growth of the aluminum phase. However, the acceleration period is difficult to control. Thus, how to decelerate the decomposition rate by prolonging the induction period through modifying the surface components of AlH$_3$ is the next research direction.

3.  The stabilization of AlH$_3$ involves enhancing crystal purity, isolating from external stimuli, and eliminating factors contributing to instability. Among the stabilization methods, surface passivation and the doping method will cause energy loss and negatively affect the combustion performance. As for surface coating, the ratio and thickness of the coating layer proves difficult to control, resulting in failure to accurately regulate its impact on the energy performance of $\alpha$-AlH$_3$. Efficient coating materials are crucial for advancing the application of AlH$_3$ in solid propellants.

**Author Contributions:** Writing—original draft preparation, Y.L.; writing—review and editing, F.Y.; investigation and supervision, Z.W. and Z.Z.; data curation and funding acquisition, Y.Z. All authors have read and agreed to the published version of the manuscript.

**Funding:** This research received no external funding.

**Data Availability Statement:** No new data were created or analyzed in this study. Data sharing is not applicable to this article.

**Conflicts of Interest:** Author Yang Zhang was employed by the company Xi'an Modern Chemistry Research Institute. The remaining authors declare that the research was conducted in the absence of any commercial or financial relationships that could be construed as a potential conflict of interest.

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
