# Peer review of "AlH3 as High-Energy Fuels for Solid Propellants: Synthesis, Thermodynamics, Kinetics, and Stabilization"

_compounds, doi:10.3390/compounds4020012_

Round 1
Reviewer 1 Report
Comments and Suggestions for Authors
Dear Authors,
Developing methods for synthesizing stable hydride compounds, studying their thermodynamic, kinetic, and energetic properties, and contributing new data to the database of chemical compounds are current challenges. Researchers are exploring ways to improve the reversibility of aluminum hydride (AlH3) dehydrogenation to make it a more efficient and viable option for hydrogen storage. Understanding these features of AlH3 is critical to developing systems that can efficiently store hydrogen and release it when needed, as well as utilize it as a high-energy fuel in different compositions.
There are several questions and remarks about the work (see attached file).

Reviewer 2 Report
Comments and Suggestions for Authors
Aluminum hydride has attracted a lot of attention due to its huge energetic properties and exceptionally high hydrogen density. In literature, one can find several papers on the perspective of the use of AlH3 as energy storage materials like paper: Graetz, J., Reilly, J. J., Yartys, V. A., Maehlen, J. P., Bulychev, B. M., Antonov, V. E., Gabis, I. E. Aluminum hydride as a hydrogen and energy storage material: Past, present and future. Journal of Alloys and Compounds, 509, S517–S528 (2011).
It is also commonly known that this material has been for years used as a propellant for rockets both for military and space exploration purposes. The authors of this paper tried to review all available physical and chemical properties of ALH3 that can be useful for practical application as solid fuel propellants. To make this review complete some available information should be added like the synthesis of AlH3 from elements and the high-pressure influence on its structure and physical properties. Implementation of these suggestions in my opinion could improve the manuscript and make it useful for the readers. So my recommendation is: to publish with minor corrections.
Reviewer 3 Report
Comments and Suggestions for Authors
Instructive paper for a non specialist. Some improvement of the language is suggested. Maybe that can be made by the journal.
Comments on the Quality of English LanguageSome improvement of the language is suggested. Maybe that can be made by the journal.
Author Response
Thank you for your valuable suggestions. We have checked the manuscript thoroughly to find possible syntax errors and fixed sentences which were too long and complex.
Reviewer 4 Report
Comments and Suggestions for Authors
AlH3 is deemed as one of the most promising high-energy fuels for solid propellants due to its high volumetric hydrogen capacity. However, AlH3 is not sufficiently stable. In this manuscript “AlH3 as High-Energy Fuels for Solid Propellants: Synthesis, Thermodynamics, Kinetics and Stabilization”, the authors summarize the synthesis and stabilization of AlH3, and then explain the instability from thermodynamics and kinetics. The manuscript is well designed. However, there are some problems in the present manuscript.
Question 1: Introduction needs more careful edited, for example “Firstly, section 1 presents the synthesis methods of aluminum hydride”, in fact, the synthesis methods of aluminum hydride is in section 2.
Question 2: There are some tense issues in manuscript, such as Line 220 “The results indicated that the samples are stable below 150 °C”, please check the tense throughout the entire text.
Comments on the Quality of English LanguagePlease check some spelling errors.
Reviewer 5 Report
Comments and Suggestions for Authors
Good morning, I read your manuscript with great interest. The presented issues are still relevant, and the summary and commentary on Synthesis, Thermodynamics, Kinetics and Stabilization of AlH3 are very well done. The chapters of the work are arranged logically and are not difficult to understand. This applies to both the substantive, editorial and linguistic aspects. A well-selected set of references. In sum: I do not notice any significant shortcomings in the manuscript.
Author Response
Thank you for your affirmation of this article